# Development of anatomically accurate digital organ models for surgical simulation and training

**Takashi Kimura**[1]*, **Kazuaki Takiguchi**[2], **Shigeyuki Tsukita**[1], **Makoto Muto**[1], **Hiroto Chiba**[1], **Naoya Sato**[1], **Yasuhide Kofunato**[1], **Teruhide Ishigame**[1], **Akira Kenjo**[1], **Hideaki Tanaka**[2], **Shigeru Marubashi**[1]

1 Department of Hepato-Biliary-Pancreatic and Transplant Surgery, Fukushima Medical University, Fukushima-city, Fukushima, Japan, 2 Department of Pediatric Surgery, Fukushima Medical University, Fukushima-city, Fukushima, Japan

* tkimura@fmu.ac.jp

## Abstract

Advancements in robotics and other technological innovations have accelerated the development of surgical procedures, increasing the demand for training environments that accurately replicate human anatomy. This study developed a system that utilizes the AutoSegmentator extension of 3D Slicer, based on nnU-Net, a state-of-the-art deep learning framework for automatic organ extraction, to import automatically extracted organ surface data into CAD software along with original DICOM-derived images. This system allows medical experts to manually refine the automatically extracted data, making it more accurate and closer to the ideal dataset. First, Python programming is used to automatically generate and save JPEG-format image data from DICOM data for display in Blender. Next, DICOM data imported into 3D Slicer is processed by AutoSegmentator to extract surface data of 104 organs in bulk, which is then exported in STL format. In Blender, a custom-developed Python script aligns the image data and organ surface data within the same 3D space, ensuring accurate spatial coordinates. By using Blender's CAD functionality within this space, the automatically extracted organ boundaries can be manually adjusted based on the image data, resulting in more precise organ surface data. Additionally, organs and blood vessels that cannot be automatically extracted can be newly created and added by referencing the image data. Through this process, a comprehensive anatomical dataset encompassing all required organs and blood vessels can be constructed. The dataset created with this system is easily customizable and can be applied to various surgical simulations, including 3D-printed simulators, hybrid simulators that incorporate animal organs, and surgical simulators utilizing augmented reality (AR). Furthermore, this system is built entirely using open-source, free software, providing high reproducibility, flexibility, and accessibility. By using this system, medical professionals can actively participate in the design and data processing of surgical simulation systems, leading to shorter development times and reduced costs.

**Data availability statement:** All python script files are available from the https://github.com/tk1971-Jpn.

**Funding:** TK: JSPS KAKENHI Grant Number JP19K09100 Japan Society of Promotion of Science(JSPS) https://www.jsps.go.jp The funders had no role in study design, data collection and analysis, decision to publish, or preparation of the manuscript.

**Competing interests:** The authors have declared that no competing interests exist.

## Introduction

The emergence of new technologies, including robotics, has led to rapid advancements in surgical procedures [1,2]. While there is increasing anticipation for applying these technologies to more complex areas [3], it is crucial to develop equipment and provide training in environments that accurately replicate human anatomy. Cadaver training offers an ideal environment, but its use is limited by ethical issues, the need for specialized techniques for handling and preservation, and the requirement for dedicated facilities [4,5].

Mixed Reality (MR) and Augmented Reality (AR) offer innovative approaches to enhancing the learning experience of surgical residents. These technologies provide immersive environments that replicate real surgical scenarios, allowing trainees to practice and refine their skills in a controlled, risk-free setting. However, integrating these technologies into surgical training presents several challenges. While AR and MR provide a high level of realism, creating simulations that accurately replicate the tactile and visual aspects of surgery remains a challenge. The cost of implementing high-precision AR and MR systems is a major barrier [6–9].

If a system utilizing digital technology could be developed to accurately reproduce human anatomy from clinical imaging data, it would significantly contribute to the creation of surgical training environments. With advancements in artificial intelligence and deep learning, technologies for automatically segmenting specific organs from CT scans and MRI and reconstructing them in 3D have significantly progressed. However, for many organs, the accuracy is still insufficient [10–12].

In this study, we developed a method in which the automatically extracted surface data of organs, along with the original imaging data, are placed within CAD software based on precise spatial coordinates. This approach allows for the editing, refinement, or reconstruction of parts that were not fully captured through automatic extraction by referencing the imaging data. As a result, we have developed a system capable of creating comprehensive digital anatomical content that includes all necessary anatomical features.

Furthermore, this paper introduces various applications of the data created using this system, including the development of surgical simulators using 3D printing, the application of hybrid simulators using animal organs, and the potential integration of augmented reality (AR) to enhance training experiences.

## Materials and methods

### Software, programming, and computer

Surface data for the organs were created with Blender version 4.3.2 for Mac (Blender Foundation, https://www.blender.org/). Automatic extraction of surface data from digital imaging and communications in medicine (DICOM) format data was performed with 3D-Slicer version 5.6.2 (https://www.slicer.org/) [13] and Horos version 3.3.6 (Horos Project, https://horosproject.org/) and saved as standard triangulated language (STL) format data. The creation of the multi planar reconstruction (MPR) image in joint photographic experts group (JPEG) format from the DICOM and the placement of the MPR image in Blender were partially automated using Python scripts. This study was conducted on a computer equipped with an Apple M2 Max chip and 64GB of memory.

### Automatic creation of JPEG format MPR images from DICOM data using Python script

Multi planar reconstruction (MPR) images in JPEG format were created using Python scripts with Pydicom, NumPy, Matplotlib, sys, and glob. The Python script was run on Jupyter

Notebook. A 3D array of CT values was created from DICOM data using NumPy. Images were then created from the 3D arrays using Matplotlib, resolution, window level, and window width were set, and saved in JPEG format. Script can be obtained from the following URL: https://github.com/tk1971-Jpn/DICOM-to-JPEG

To create the script, we used some of the information from the following web page: https://Pydicom.github.io/pydicom/dev/auto_examples/image_processing/reslice.html

### Auto segmentation of organ

The organ surface data were generated using the TotalSegmentator extension [14,15] in 3D-Slicer and then exported in STL format for most organs. The body surface data were created with Horos and exported in STL format. (Please refer to the step-by-step protocol for details.)

### Implementing a MPR image slider in Blender using Python

We created a slider interface using Blender's bpy.props.IntProperty, which allowed users to move through the MPR images interactively. The slider's range was set dynamically based on the total number of DICOM slices, and every time the slider value changed, the display was updated to show the corresponding slice image. Note that the size of the original human body image is far beyond the scope of Blender's workspace, so it is necessary to reduce the size of the original image to 1/10–1/50 of its original size when placing it.

The script to run slider within Blender can be obtained from the following URL: https://github.com/tk1971-Jpn/Slider-viewer-in-Blender

### Importing organ surface data into Blender

The surface data was imported into Blender collectively using the script below. During import, object names were automatically changed to the corresponding organ names. Additionally, with the creation of a surgical simulator in mind, the data was positioned in Blender as if in the supine position, with the head-tail direction aligned with the y-axis and the anterior-posterior direction aligned with the z-axis.

The script to run within Blender can be obtained from the following URL: https://github.com/tk1971-Jpn/Import-organ-STL-data-into-Blender

### Design and practical application of a surgical simulator

A metal pedestal was created to reproduce the spatial coordinates in Blender in the real world (NAKANO INC., Fukushima). This pedestal has four different lengths (5 cm, 10 cm, 20 cm, and 30 cm) of 1.5mm cylindrical posts that can be placed in holes drilled at 3 cm intervals in the plate. Since the data for this pedestal is also located in the Blender workspace, a simulator that reproduces the distances and angles in 3D space can be created and put to practical use by designing the simulator to be attached to the pedestal.

The digital data of the metal base can be obtained from the following https://github.com/tk1971-Jpn/Simulator-base.

### Acquisition of DICOM data

Data for the cases used in this study were obtained from The Cancer imaging archive at the NIH (TCIA, https://dev.cancerimagingarchive.net).

### Augmented reality

The data for AR was created by exporting objects made in Blender as Universal Scene Description (.usd) files, which were then converted to .usdz format using Reality Converter

(Apple Inc, https://developer.apple.com/jp/augmented-reality/tools/). AR was executed on an iPhone or iPad using Reality Composer (Apple Inc).

### Ethical consideration

This study was conducted with the approval of the Ethics Committee of Fukushima Medical University (Approval number: general 30165).

In this study, stored imaging data from patients who underwent diagnostic imaging at Fukushima Medical University between 01-06-2019, and 31-07-2020, were used. Anonymization was performed when extracting the data from the electronic medical record system. Patient names, ages, and examination dates were all replaced with dummy data. Patient consent for the use of data was obtained through an opt-out method: the research plan was published on the university's website, and requests to refuse the use of information were accepted. Any data subject to such requests were excluded from the study.

In the preparation of this manuscript, we used public data obtained from TCIA. Therefore, no consent was obtained for the use of this data.

### Step-by-Step protocol

The protocol described in this peer-reviewed article is published on protocols.io, https://dx.doi.org/10.17504/protocols.io.bp2l6d19dvqe/v4 and is included for printing as S1 File with this article.

### Detailed demonstration

A detailed demonstration of the workflow published in this study, using TCIA data, is available at the following URL.

https://github.com/tk1971-Jpn/Demonstration-of-Organ-data-set-creation

## Result

### Outline of the research

Fig 1 provides an overview of this study. From the DICOM data of contrast-enhanced CT scans, MPR images in JPEG format and STL surface data of organs segmented automatically in 3D-Slicer and Horos are created and arranged in Blender. In Blender, the imported surface data is modified, and organs not captured by automatic segmentation are created to construct a digital anatomical dataset that includes all necessary organs. Based on this dataset, a surgical training simulator faithfully reproducing human anatomy in a virtual space is designed, which can be used for simulation in virtual space and training in physical space by materializing it with a 3D printer. Furthermore, applications for augmented reality, combining these elements, are also pursued.

### Creation of MPR images from DICOM data using Python script

Since DICOM data cannot be directly imported into Blender, it is first converted into JPEG-format MPR images, saved, and then referenced for display within Blender. Fig 2 presents an overview of the process for generating MPR images from DICOM data. The Python libraries utilized in this process include Pydicom, NumPy, Matplotlib, sys, and glob. The program was executed on Jupyter Notebook for each cell, and information contained in DIOCM data was obtained in each process. Initially, DICOM data is converted into a 3-dimensional NumPy array. The grayscale is adjusted along the way, and MPR images are then generated from this array using Matplotlib, named, and stored in a predefined directory.

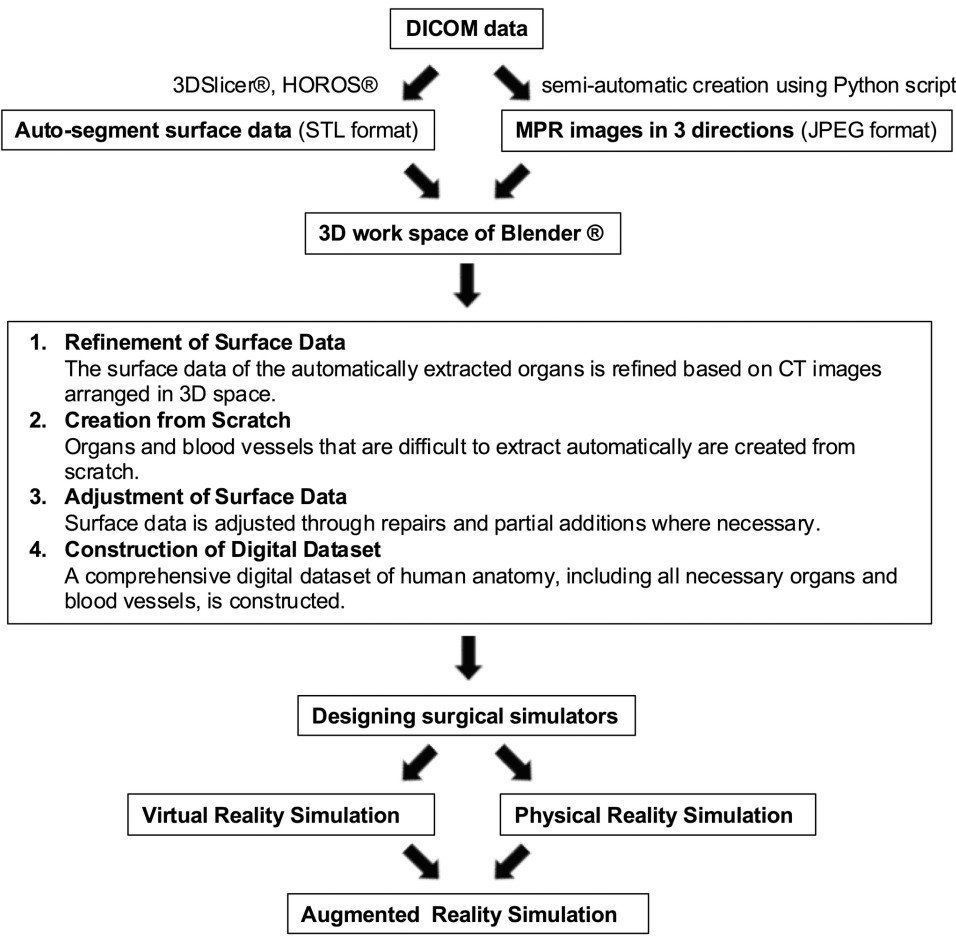

**Fig 1. Outline of the research.**

## Setting up a work environment for constructing a comprehensive organ surface dataset in Blender

In advance, the organ surface data is segmented from DICOM data using the AutoSegmentator extension in 3D-Slicer and Horos, and then saved in STL format in the specified folder (For details on how to use 3D-Slicer and Horos, please refer to the S1 File).

MPR images were set up in Blender, allowing specific images to be displayed using sliders implemented through a script (Fig 3A and 3B).

Next, STL data of organs automatically segmented in 3D-Slicer and body surface (skin) data segmented in Horos were imported and aligned based on the displayed CT images (Fig 3C). With the creation of a surgical simulator in mind, each organ was positioned in the supine orientation. Furthermore, since the size of the human body greatly exceeds the workspace in Blender, it was necessary to reduce the scale upon import: to 1/50 for adults and to 1/10 for pediatric cases.

To evaluate the efficiency of this system, data was obtained from 12 different The Cancer Genome Atlas (TCGA) projects in TCIA, and the time required for data processing from DICOM data to the state corresponding to Fig 3C was measured (S2 File and S1 Fig). When using the Fast mode of 3D-Slicer (which generates slightly coarse surface data), the average processing time was approximately 7 minutes, while using the Full-resolution mode resulted

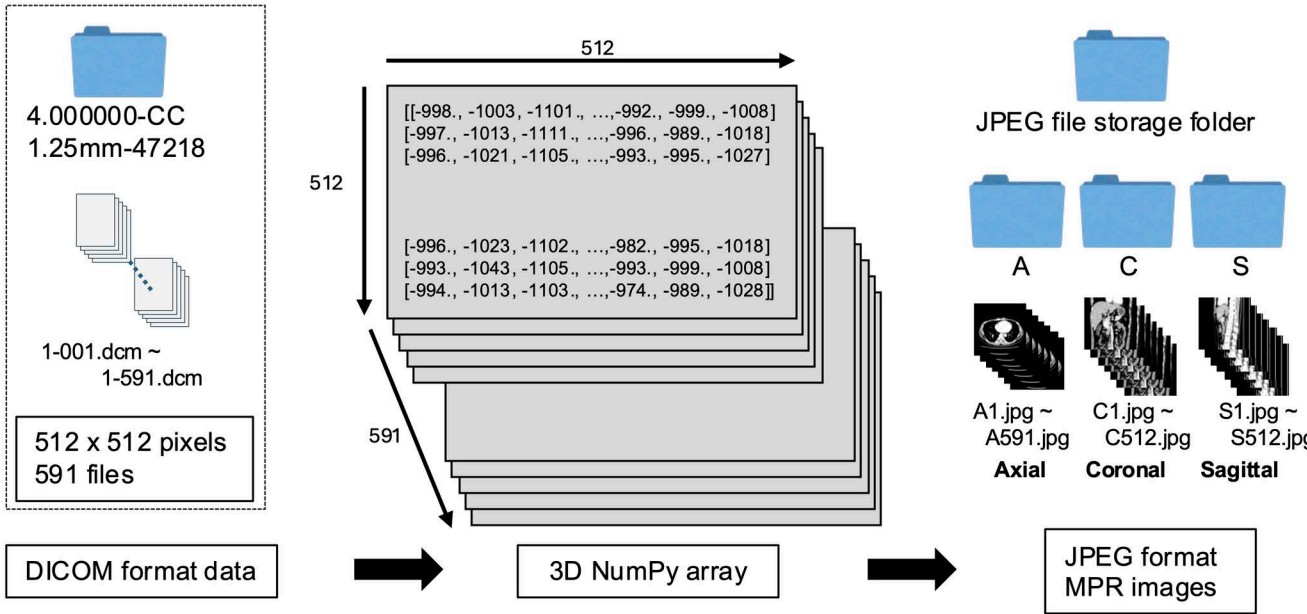

**Fig 2. Generation of MPR images in JPEG format from DICOM data.** (A) In the example used, DICOM data is stored as 512 × 512 pixel images across 591 axial sections in the cephalocaudal direction. (B) Converting DICOM data to a 3D NumPy array. (C) Generating MPR images in JPEG format from a NumPy array and save it in the specified folder.

in an average processing time of approximately 22 minutes. Regardless of the number of data points, imaging intervals, or scanned regions, all data successfully achieved the intended purpose. In TCIA, patient data such as height and weight are not publicly available. Therefore, numerical assessments cannot be conducted. However, based on the obtained data images, the cases range from lean to severely obese, and it was confirmed that the surface data of the target organ could be successfully acquired (S1 Fig).

The surface data of each imported organ was processed based on CT images, with noise removed, surfaces smoothed, and shapes adjusted. In Blender, shapes can be modified by moving vertices, edges, and faces in modeling mode (Fig 3D), or sculpted like clay in sculpting mode (Fig 3E). Furthermore, by importing spherical objects and deforming them to match organ boundaries, or by aligning linear objects with the course of blood vessels, organs and blood vessels that were not automatically extracted can be newly created and added by referring to the CT images (Fig 3F).

As shown in Fig 4A–4D, the data obtained through automatic extraction cover most of the target organ but do not include detailed anatomical structures such as peripheral blood vessels, the common bile duct, and the ureter. These structures were created from scratch by referencing the CT images. Additionally, by connecting disconnected blood vessels and refining organ boundaries with reference to the CT images, a more realistic digital dataset of the organs could be constructed (Fig 4E). Through these processes, a digital anatomical dataset was constructed that faithfully reproduces the necessary organs from the contrast-enhanced CT data (Fig 4F).

## Application of organ 3D surface data to surgical simulation

The dataset created using this method can be edited in Blender and converted into digital content tailored to various purposes. Additionally, since it can be exported in multiple file

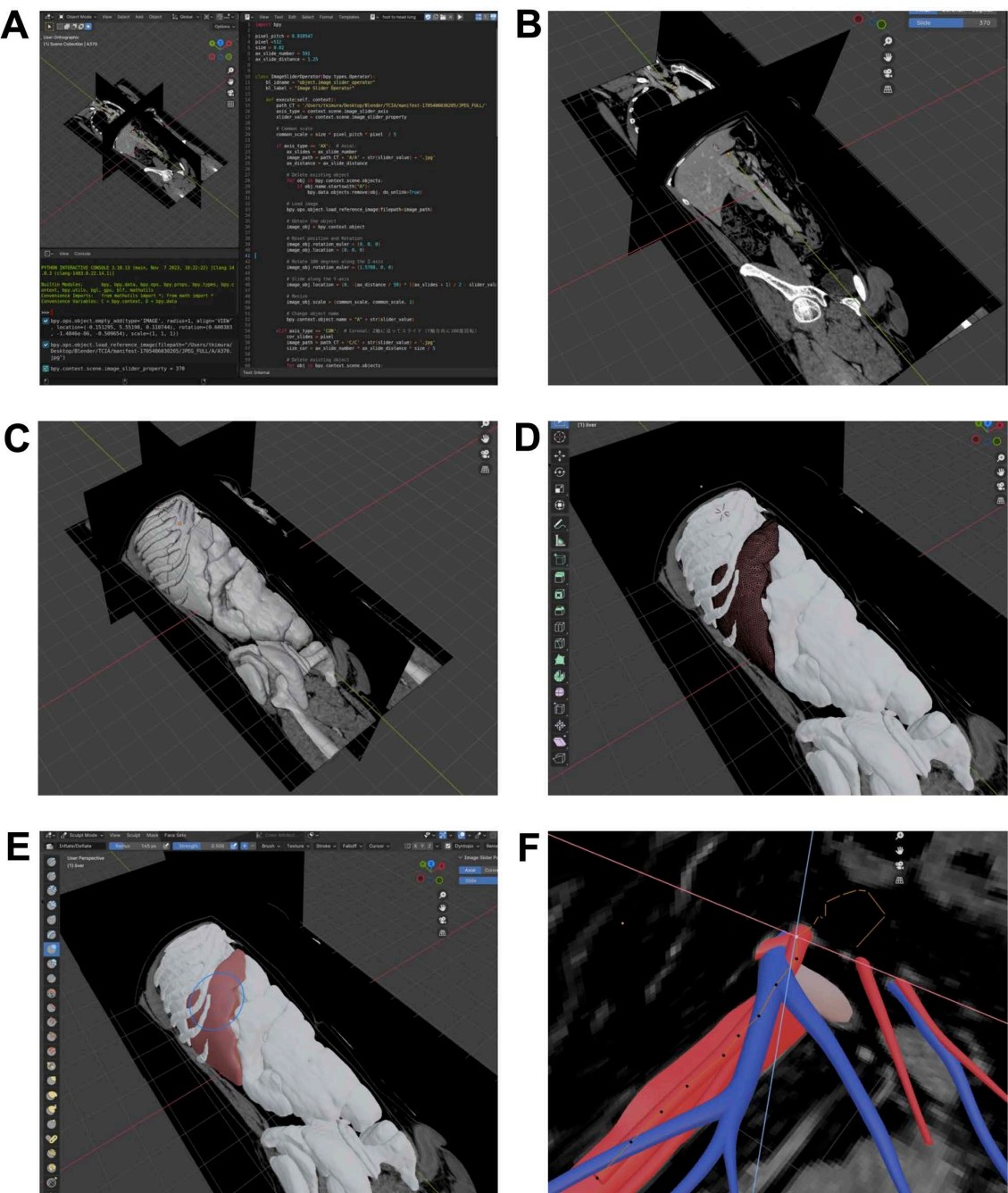

**Fig 3. Construction of a comprehensive organ surface dataset in Blender.** (A) Display any MPR image using sliders by executing a script in the script window. (B) By moving the slider in the upper right of the screen, any image can be displayed in the three axial directions. (C) STL format organ data is imported in bulk. (D) In Blender's modeling mode, shapes are modified by moving vertices, edges, and faces. (E) In Blender's Sculpting mode, shapes are adjusted as if molding clay. (F) Creating blood vessels from scratch based on CT images for those that could not be automatically extracted.

formats, its application in new software is expected to expand its usability. Several examples of how this dataset can be utilized are presented below.

**Creation of a surgical simulator using 3D Printing.** Fig 5 illustrates the process of creating a simulator for training in laparoscopic choledochojejunostomy. First, the initial

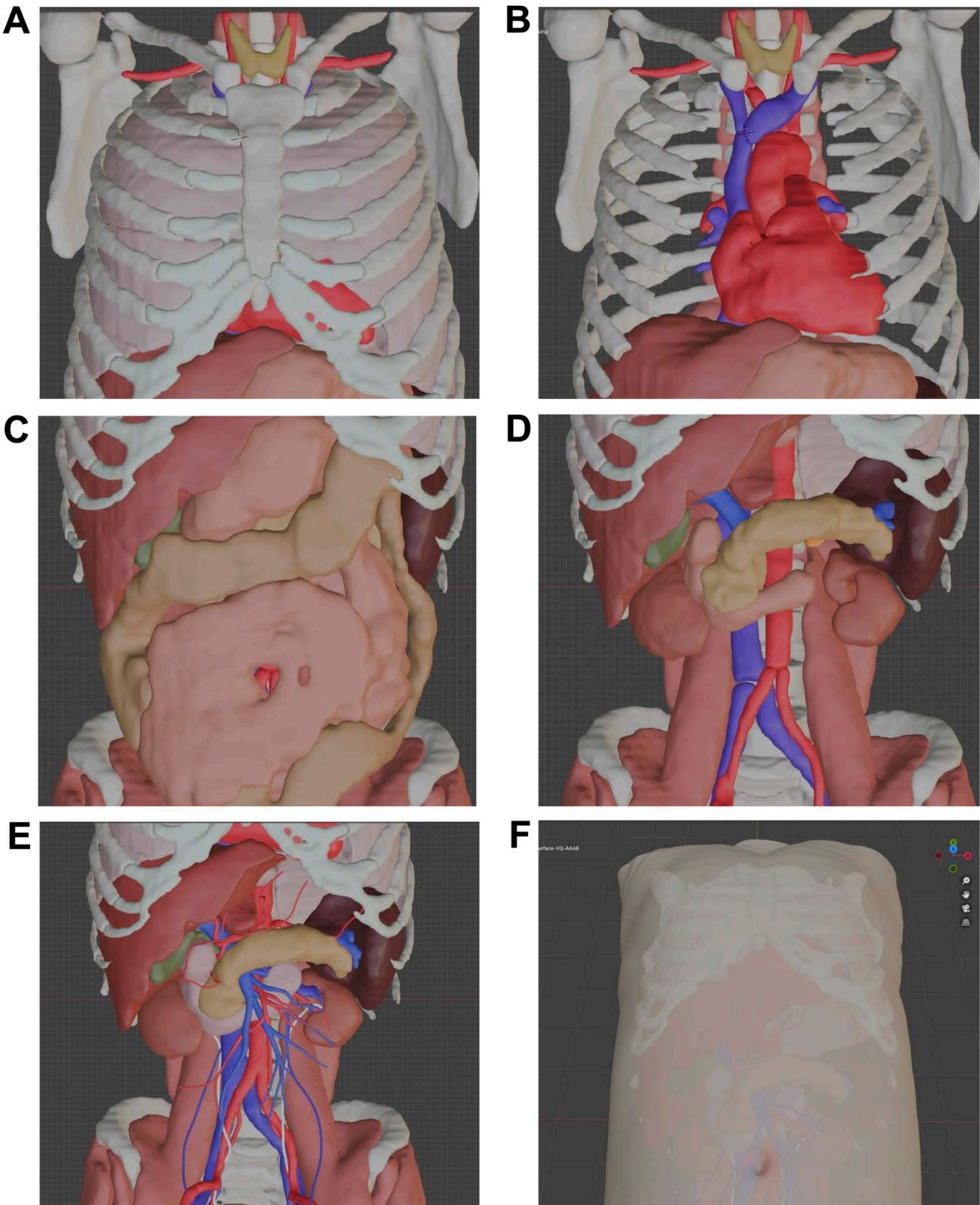

**Fig 4. A Representative example of organ surface data before and after editing.** (A) All thoracic organs (adjust mesh count of data and set colors). (B) Thoracic organs (with lungs and costal cartilage hidden). (C) All abdominal organs (adjust mesh count of the data and set colors). (D) Abdominal organs (with the stomach, small intestine, and colon hidden). (E) By adjusting the organ surfaces to align with the CT images, joining separately extracted blood vessels, and adding peripheral vessels (arteries, portal veins, veins), the common bile duct, and ureters, a more realistic digital dataset of the organs can be constructed. (F) Body surface (displayed with increased transparency to allow visualization of the organs).

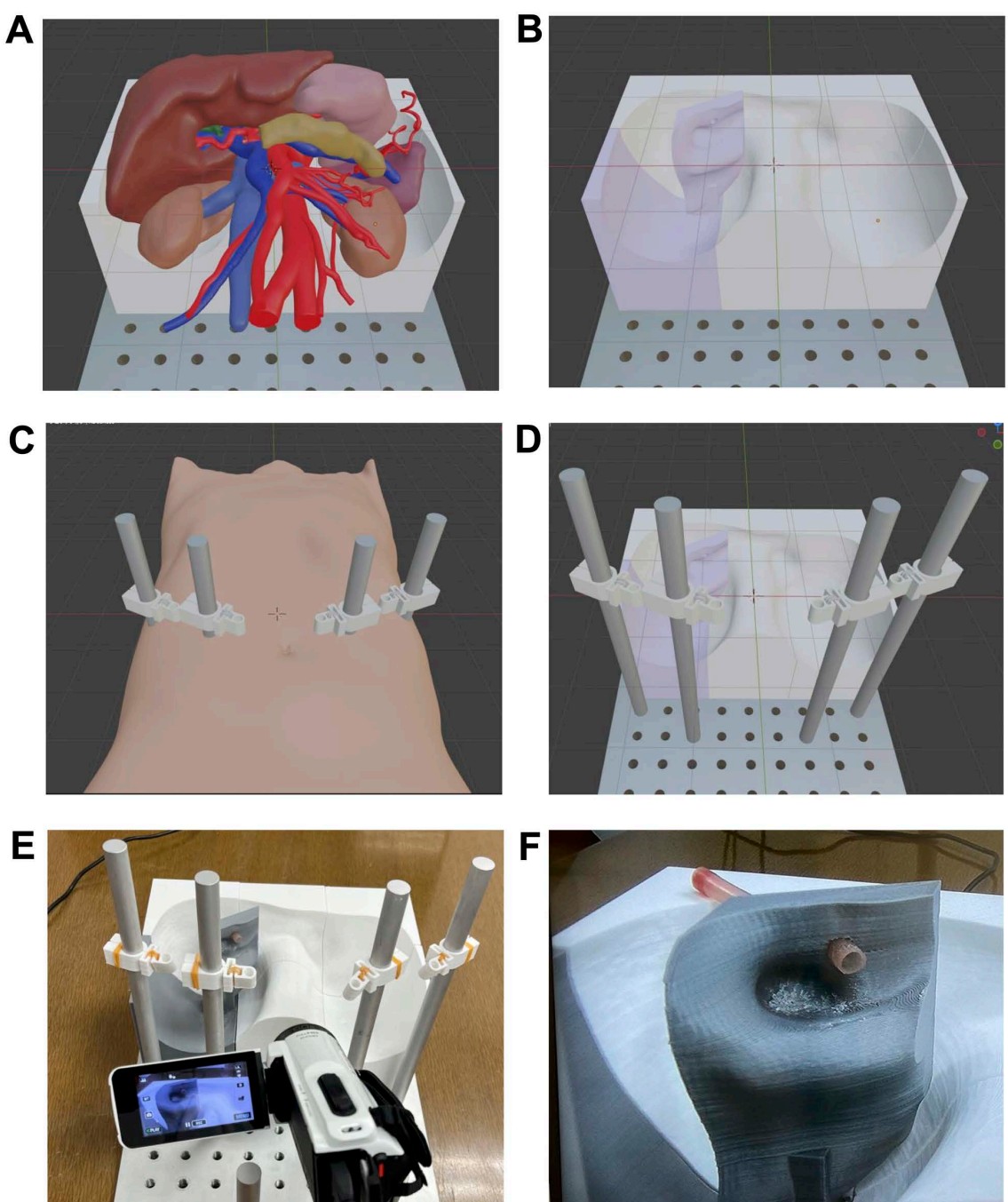

**Fig 5. Creation of a surgical simulator using 3D printer.** Process of creating a simulator for training in laparoscopic choledoco-jejunostomy. (A) Scene at the start of training in a virtual space (B) Hepatic hilum for bile duct placement (purple). (C) Determine the port positions based on skin data (configured with consideration for the position during pneumoperitoneum). (D) A simulator designed in virtual space. (E) A simulator materialized by 3D printing (F) The surgical target area projected on the monitor.

scene at the start of the training is reproduced in a virtual space. Next, the necessary components for performing the procedure are designed to be minimal while maintaining their spatial coordinates. The positions where the surgical forceps pass through are set to preserve spatial coordinates. This ensures that the distance and angle from the insertion point of the forceps to the target organ are accurately replicated.

**Hybrid simulator using animal organs.** Fig 6 illustrates the process of creating a hybrid simulator using animal organs. In surgical training, in addition to spatial coordinates, factors such as the stiffness of the target organ and its fine-layered structure are also crucial.

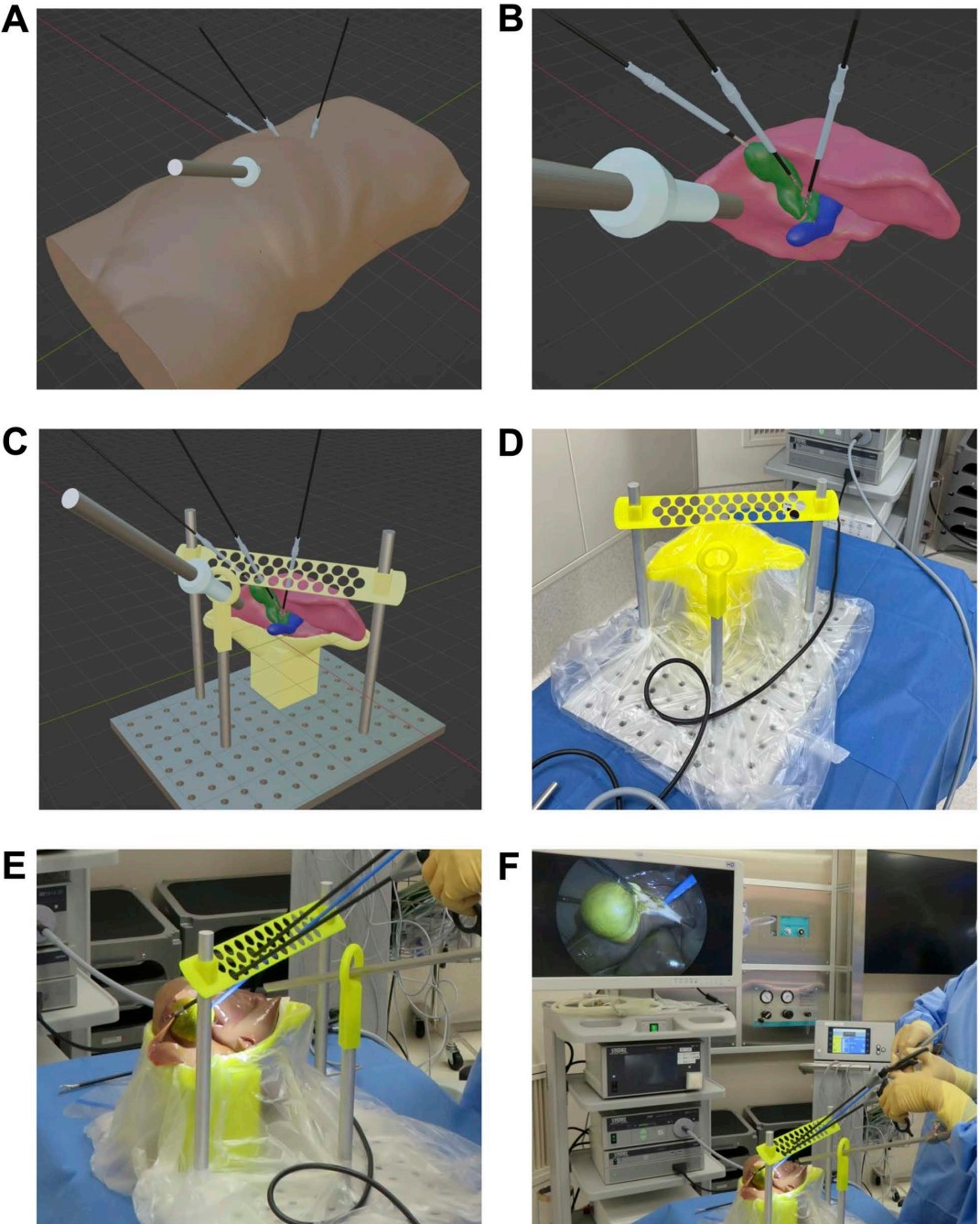

**Fig 6. Hybrid simulator using animal organs.** Process of creating hybrid simulation system for training in laparoscopic cholecystectomy using animal organs. (A) Identification of forceps and camera insertion positions in a virtual space. (B) Reproduction of surgical scenes in a virtual space. (C) Designing a platform for placing the liver and holes for forceps and camera passage. (D) A simulator created by 3D printing. (E) A scene of training with a porcine liver placed in the simulator. (F) A scene in the operating room.

By positioning animal organs, such as those from pigs, in anatomically correct locations corresponding to the human body, a training system that closely replicates real surgical procedures can be constructed. This example presents the method for creating a simulator for laparoscopic cholecystectomy. First, the surgical procedure is recreated in a virtual space (Fig 6A and 6B), and a platform for positioning the liver, as well as the locations where the forceps and camera pass through, are designed (Fig 6C). By placing a liver in the simulator produced by 3D printing and conducting training (Fig 6D and 6E), a realistic surgical experience can be achieved (Fig 6F).

**Application of augmented reality to surgical simulation.** Fig 7 presents an application example of augmented reality. First, the entire simulator is designed in a virtual space (Fig 7A). Then some parts of the simulator are materialized using 3D printing (Fig 7B), while others are converted into VR content. The VR content is exported as USD format data and then converted to USDZ using dedicated software (Fig 7C). To set the projection position of the VR content, anchors are created, and adjustments are made to align the virtual anchor with the real anchor (Fig 7D and 7E), ensuring accurate VR projection placement. By using a smartphone or tablet, VR images can be projected onto real objects (Fig 7F and 7G).

## Discussion

This study developed a method for constructing more accurate anatomical datasets by utilizing open-source software and free 3D CAD tools to refine and supplement incomplete organ surface data automatically extracted from DICOM data by AI and deep learning, using the original images as a reference. The data generated using this method accurately reproduces the spatial coordinates of organs, making it a potentially applicable dataset for creating various simulators used in real-world training.

Additionally, by utilizing widely used free programming languages and software such as Python and Blender, this study has provided a method that anyone can use without incurring introduction costs.

Python is widely used in fields such as data analysis and deep learning and is well-known in the medical field. As an object-oriented programming language, it is easy to understand even for non-specialists and can be learned relatively quickly [16]. Furthermore, with the recent advancements in AI-assisted programming, practical implementation can now be achieved in a shorter time [17,18].

Blender, originally developed as 3D graphics software for animation production, possesses advanced digital content creation capabilities and is suitable for developing surgical simulators. Numerous reports have already documented the use of Blender for surgical training simulators [19–23]. Furthermore, Blender can be operated using Python scripts, enabling complex operations to be performed programmatically in a single batch, significantly improving work efficiency.

While automatic segmentation technology has made significant progress for many organs, challenges still remain for certain structures, particularly complex organs like the pancreas, where recent methods such as template-based surface extraction using spherical topology achieve only moderate Dice Similarity Index values [24,25]. Accurately correcting incomplete organ surface data extracted automatically has been considered difficult. However, we developed a method to import such data along with CT images into the same 3D space within CAD software, enabling precise corrections by referencing the images. Furthermore, for organs that cannot be extracted automatically, it is also possible to create them from scratch within the same 3D space by referring to the image data. Since there is no gold standard for evaluation, the accuracy of the data generated by this system cannot be strictly measured. However, considering that additional processing has been applied to the automatically extracted data

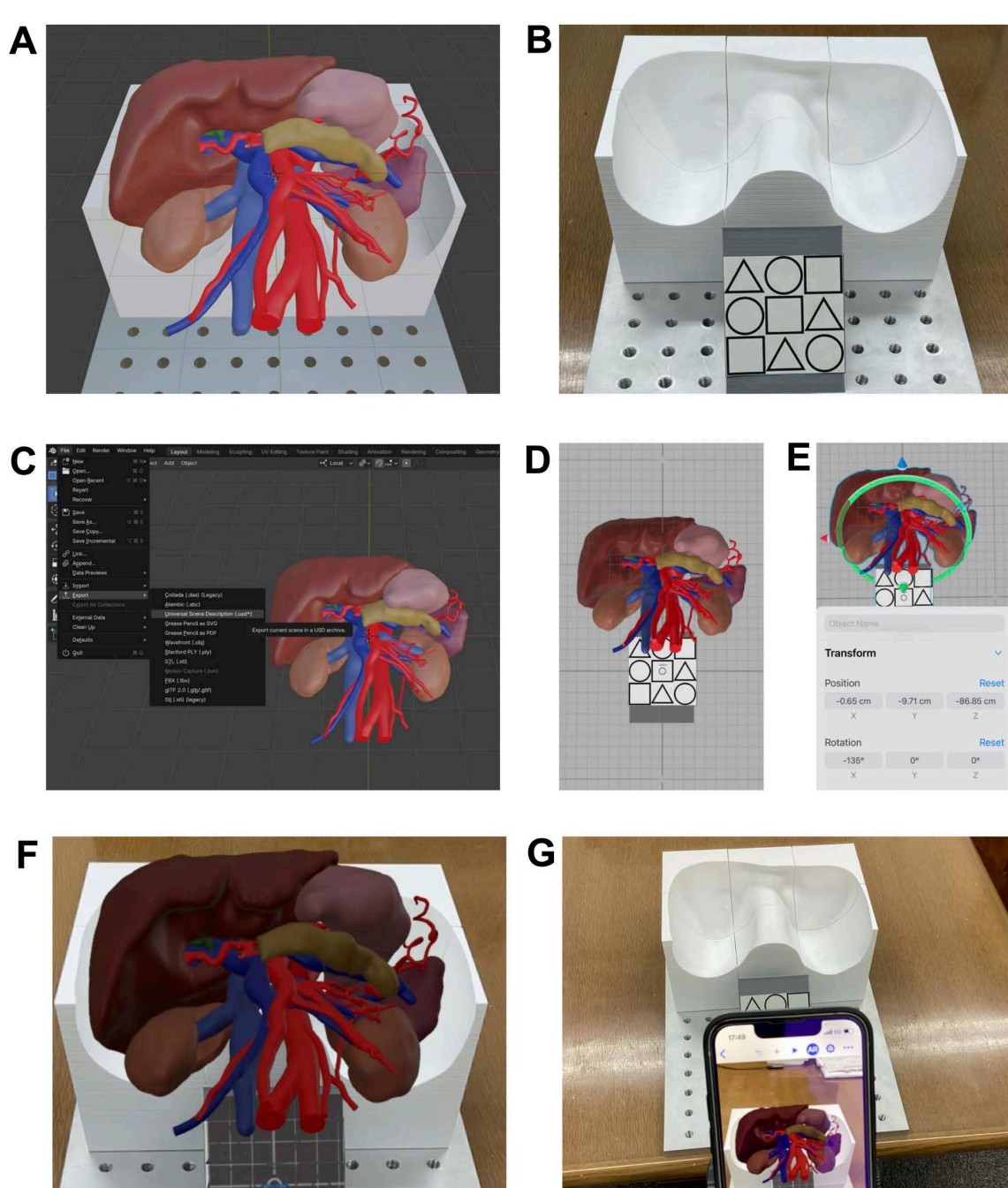

**Fig 7. Application of augmented reality to the creation of surgical simulators.** (A) Designing organs and the abdominal cavity in a virtual space. (B) Materializing only the abdominal cavity using 3D printing. (C) Acquisition of USD format data. (D) Setting an anchor (E) Adjust the position of the virtual space object so that the anchor in the virtual space aligns with the anchor in the real world. (F) An image combining the real world and the VR space displayed on a smartphone monitor. (G) A scene where a VR object is projected onto a real-world object using a smartphone and AR technology.

using the highest level protocol currently available, nnU-Net [25], as adopted by 3D-Slicer's AutoSegmentator, to bring it closer to the original image data, it can be said that the results are approaching a more ideal outcome.

In this study, we also presented the potential applications of the anatomical digital dataset created using this system. This dataset can contribute to the development of simulators that accurately reproduce spatial coordinates, which are essential for training in laparoscopic and robot-assisted surgery. The angle and distance from the port insertion site to the target organ for the procedure can be faithfully replicated. The metal base introduced in this study plays a crucial role in reproducing the spatial coordinates of a VR-designed simulator in the real world.

This system is based on the subtractive approach, which involves removing unnecessary elements from a maximally prepared dataset. While the subtractive approach entails a high initial implementation cost, it enables more efficient long-term operation by facilitating customization, shortening data processing time, and ensuring the extraction of the minimal necessary data [26]. The data processing system developed in this study can significantly contribute to reducing this initial implementation cost.

In surgical simulators, factors such as tactile feedback, organ stiffness, and texture are important considerations. While the development of synthetic organs is actively progressing, it is still in the process of optimization, including issues related to manufacturing costs [27,28]. As one possible approach to this issue, surgical simulation using animal organs is also actively conducted [29,30]. In this study, we presented an example of a simulator that places animal organs in anatomically correct human positions. Although this type of simulator requires facilities that comply with animal training regulations due to biohazard concerns, it offers a training experience that closely mimics human tissue, unlike dry lab models. Moreover, porcine organs are relatively inexpensive and readily available, making this setup particularly effective for repetitive training sessions.

AR is one of the most actively researched fields in surgical training [31,32]. By projecting VR images onto physical models in real space, AR enables the creation of an immersive training environment with tactile feedback. As demonstrated in this study, free-to-use applications are also becoming widely available, further promoting the adoption of AR technology. The organ dataset created using the system developed in this study can be easily converted into AR content for practical use.

The creation of surgical simulators requires collaboration not only among medical professionals but also with experts in computing, manufacturing, and other technical fields. In the system developed in this study, surgeons in need of training can actively participate in the development process, allowing them to leverage their medical expertise and accurately reflect their specific requirements. This approach is expected to reduce costs and shorten development timelines.

## Conclusions

We developed a system that utilizes open-source, free software to modify and refine automatically extracted organ surface data within CAD software, using reference images to achieve datasets that more closely match the ideal anatomical structures. This system enables the creation of high-quality datasets that can be applied to a wide range of surgical simulations, offering significant potential for enhancing training, planning, and research in the field of surgical education.

## Supporting information

**S1 File. Step by step protocol.** Step by step protocol, also available on protocols.io. (PDF)

**S2 File. Time required to import and arrange organ surface datasets and MPR images from TCIA in Blender.** The time required to generate MPR images in JPEG format from

DICOM data and to extract organ surface data using AutoSegmentator in 3D Slicer was measured using data from 12 cases obtained from TCIA.
(XLSX)

**S1 Fig. The organ surface data in Blender.** The organ surface data from 12 TCIA cases, displayed in Blender along with the MPR images.
(TIF)

## Author contributions

**Conceptualization:** Takashi Kimura, Hideaki Tanaka, Shigeru Marubashi.

**Data curation:** Takashi Kimura, Kazuaki Takiguchi, Shigeyuki Tsukita, Makoto Muto, Hiroto Chiba, Naoya Sato, Yasuhide Kofunato, Teruhide Ishigame.

**Formal analysis:** Kazuaki Takiguchi, Makoto Muto, Naoya Sato, Yasuhide Kofunato, Teruhide Ishigame.

**Funding acquisition:** Takashi Kimura, Shigeru Marubashi.

**Methodology:** Takashi Kimura, Naoya Sato, Teruhide Ishigame, Akira Kenjo.

**Software:** Takashi Kimura, Yasuhide Kofunato.

**Supervision:** Akira Kenjo, Hideaki Tanaka, Shigeru Marubashi.

**Visualization:** Takashi Kimura, Kazuaki Takiguchi, Shigeyuki Tsukita, Makoto Muto.

**Writing – original draft:** Takashi Kimura, Hiroto Chiba, Naoya Sato, Hideaki Tanaka.

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
