## [Decision Letter · Decision Letter 0]

22 Jan 2025

Dear Dr. Kimura,

Thank you for submitting your manuscript to PLOS ONE. After careful consideration, we feel that it has merit but does not fully meet PLOS ONE’s publication criteria as it currently stands. Therefore, we invite you to submit a revised version of the manuscript that addresses the points raised during the review process.

We look forward to receiving your revised manuscript.

Kind regards,

James J Cray Jr., Ph.D.

Academic Editor

PLOS ONE

Journal Requirements:

3. We note you have not yet provided a protocols.io PDF version of your protocol and/or a protocols.io DOI. When you submit your revision, please provide a PDF version of your protocol as generated by protocols.io (the file will have the protocols.io logo in the upper right corner of the first page) as a Supporting Information file. The filename should be S1_file.pdf, and you should enter “S1 File” into the Description field. Any additional protocols should be numbered S2, S3, and so on. Please also follow the instructions for Supporting Information captions [https://journals.plos.org/plosone/s/supporting-information#loc-captions]. The title in the caption should read: “Step-by-step protocol, also available on protocols.io.”

Please assign your protocol a protocols.io DOI, if you have not already done so, and include the following line in the Materials and Methods section of your manuscript: “The protocol described in this peer-reviewed article is published on protocols.io (https://dx.doi.org/10.17504/protocols.io.[...]) and is included for printing purposes as S1 File.” You should also supply the DOI in the Protocols.io DOI field of the submission form when you submit your revision.

If you have not yet uploaded your protocol to protocols.io, you are invited to use the platform’s protocol entry service [https://www.protocols.io/we-enter-protocols] for doing so, at no charge. Through this service, the team at protocols.io will enter your protocol for you and format it in a way that takes advantage of the platform’s features. When submitting your protocol to the protocol entry service please include the customer code PLOS2022 in the Note field and indicate that your protocol is associated with a PLOS ONE Lab Protocol Submission. You should also include the title and manuscript number of your PLOS ONE submission.

“TK:

JSPS KAKENHI Grant Number JP19K09100

Japan Society of Promotion of Science(JSPS)

https://www.jsps.go.jp”

“This work was supported by JSPS KAKENHI Grant Number JP19K09100”

“TK:

JSPS KAKENHI Grant Number JP19K09100

Japan Society of Promotion of Science(JSPS)

https://www.jsps.go.jp”

6. We note that Figures 3, 4, 5 and 6 in your submission contain copyrighted images. All PLOS content is published under the Creative Commons Attribution License (CC BY 4.0), which means that the manuscript, images, and Supporting Information files will be freely available online, and any third party is permitted to access, download, copy, distribute, and use these materials in any way, even commercially, with proper attribution. For more information, see our copyright guidelines: http://journals.plos.org/plosone/s/licenses-and-copyright.

a. You may seek permission from the original copyright holder of Figures 3, 4, 5 and 6 to publish the content specifically under the CC BY 4.0 license. 

Reviewers' comments:

Reviewer's Responses to Questions

**Comments to the Author**



Reviewer #1: Yes

Reviewer #2: Yes

2. Has the protocol been described in sufficient detail?

To answer this question, please click the link to protocols.io in the Materials and Methods section of the manuscript (if a link has been provided) or consult the step-by-step protocol in the Supporting Information files.

Reviewer #1: Yes

Reviewer #2: Partly

3. Does the protocol describe a validated method?

Reviewer #1: Yes

Reviewer #2: Yes

4. If the manuscript contains new data, have the authors made this data fully available?

Reviewer #1: Yes

Reviewer #2: N/A

**5. Is the article presented in an intelligible fashion and written in standard English?**

Reviewer #1: Yes

Reviewer #2: Yes

Reviewer #1: Following a thorough review of the manuscript, the following issues and problems were identified. By addressing these issues, it is anticipated that the paper will exhibit enhanced quality and reliability, thereby more clearly demonstrating the value of the proposed system.

1. The methodology is inadequate in terms of detail.- The automatic segmentation process and the deep learning model used are not clearly explained. In order to enhance reproducibility, it is essential to provide more specific information, including the procedure, the software used, and the parameter settings.

2. Lack of evaluation metrics- Objective metrics are absent, hindering the evaluation of the effectiveness of the proposed system. For instance, the anatomical accuracy of the generated model and quantitative evaluation of training effects are required.

3. The number of samples and diversity (age, gender, body shape, etc.) of the CT data used are not clearly stated. These factors are of particular relevance given their impact on the generalisability of results.

4. Specific solutions to AR issuesThe issue of spatial consistency in AR is mentioned, but no specific approach to solving this problem or future research policy is presented.

5. Verification of cost-effectiveness and comparison with other existing methodsThe assertion that the system is "cost-effective" is not supported by specific evidence or comparative data.

A comparative analysis between the proposed system and other extant surgical training methods is absent. This would allow for a more robust demonstration of the system's superiority.

In addition, the long-term effectiveness of the system merits further investigation. The present paper does not examine the long-term effects of training using this system or its impact on actual surgical outcomes.

Reviewer #2: General Comments

Thank you for the opportunity to review “Development of Anatomically Accurate Digital Organ Models for Surgical Simulation and Training – A Methodology for Reconstructing and Applying 3D Organ Datasets in Real Space”. Overall, I think this is a worthwhile manuscript. I have a number of edits, suggestions, and comments that I would like the authors to consider before this manuscript is ready for publication.

The main comment I have for the authors about this manuscript is that while the guide that the authors created is certainly worthwhile and provides readers with a low-cost mechanism for segmentation and CAD-based design for anatomical structures, the authors overstate how easily the models created through this process could be 3D printed for surgical simulation (not to mention implementation in AR, with no mention of VR). Anatomically accurate simulation not only has to have organs and tissues that are correct size and location, but also, the texture, density, tensile strength (among many other qualities) has to be 1:1 to human tissue. Additive manufacturing practices to print these organs is evolving, and in addition, identifying the correct materials to replicate the different qualities of individual tissue types is still largely unknown (outside of a small number of tissues).

I would like the authors to update their manuscript to focus much more on the step-by-step process of their segmentation standard operating procedure to better aid readers in replicating this process simply by reading this manuscript. In addition, the authors should only mention 3D-printing or AR as mediums that could then utilize these 3D digital models with the appropriate 3D-printers, software, and materials, and dedicate software developers to incorporate the 3D models into surgical training AR (or VR) software.

Please note, all page numbers listed below are in reference to the PDF proof of the manuscript.

Introduction

This section was succinct but provided an OK background to the problem. I would like to see the authors provide an expanded background into the tangible methods of creating surgical simulation models. The vast majority of surgeons would not be able to use a 3D digitized model of an organ(s) for surgical simulation. A thorough discussion of 3D-printing and mixed reality in the world of surgery simulation would enhance this section greatly.

Page 13 line71 – please define an acronym the first time you use it (STL).

Materials and Methods

Page 15 Lines 85-88 – Did the authors begin with their DICOM file in Blender and then move it to 3D Slicer? From my experience it would be flipped. I suggest writing this section in the correct chronological order of operations.

Page 15 lines 90-101 and Page 16 lines 108-117: I am confused about the function of the multi-planar reconstruction images that are automated through a python script. Can the authors please elaborate on the function of this reconstruction images and the benefit of creating them through a script. Why is this an important step instead of simply creating the surface model that 3D Slicer can do?

AR Section – Did the authors create any type of UI or interactions for the models beyond the baseline that Apple provides with the AR Kit? Please elaborate if so.

Results

Please refer to main comment at the beginning of this review with request to update the results section to largely remove the 3D printed simulator outside of potentially showing how the authors produced their model. This is important, because while the authors 3D printed a surgical simulator based on the segmented model, there appears to be no assessment of the 3D printed simulator.

Discussion

Page 23 lines 252-254: I would like the authors to elaborate more on the creation of organs or tissues that are not auto extracted and how they could create them by simply referring to the image data.

**Do you want your identity to be public for this peer review?** For information about this choice, including consent withdrawal, please see our Privacy Policy

Reviewer #1: No

Reviewer #2: No

---

## [Author Response · Author response to Decision Letter 1]

19 Feb 2025

Response to Journal

We have made the following revisions based on your instructions. Please let us know if any further adjustments are needed.

Journal Requirements:

Response to Journal

I have revised the manuscript according to PLOS ONE's style requirements. Please let me know if any further modifications are necessary.

Response to Journal

All Python scripts used in the protocol described in this manuscript have been made publicly available in a GitHub repository. A detailed usage guide is provided in the README to ensure ease of use for readers. The main code is also made directly accessible from the protocol registered on protocol.io.

3. We note you have not yet provided a protocols.io PDF version of your protocol and/or a protocols.io DOI. When you submit your revision, please provide a PDF version of your protocol as generated by protocols.io (the file will have the protocols.iologo in the upper right corner of the first page) as a Supporting Information file. The filename should be S1_file.pdf, and you should enter “S1 File” into the Description field. Any additional protocols should be numbered S2, S3, and so on. Please also follow the instructions for Supporting Information captions [https://journals.plos.org/plosone/s/supporting-information#loc-captions]. The title in the caption should read: “Step-by-step protocol, also available on protocols.io.”

Please assign your protocol a protocols.io DOI, if you have not already done so, and include the following line in the Materials and Methods section of your manuscript: “The protocol described in this peer-reviewed article is published on protocols.io(https://dx.doi.org/10.17504/protocols.io.[...]) and is included for printing purposes as S1 File.” You should also supply the DOI in the Protocols.io DOI field of the submission form when you submit your revision.

If you have not yet uploaded your protocol to protocols.io, you are invited to use the platform’s protocol entry service [https://www.protocols.io/we-enter-protocols] for doing so, at no charge. Through this service, the team at protocols.io will enter your protocol for you and format it in a way that takes advantage of the platform’s features. When submitting your protocol to the protocol entry service please include the customer code PLOS2022 in the Note field and indicate that your protocol is associated with a PLOS ONE Lab Protocol Submission. You should also include the title and manuscript number of your PLOS ONE submission.

Response to Journal

We have provided the PDF version of protocols.io as S1_file.pdf. Additionally, we have included the protocols.io DOI in the Material and Method section of the manuscript.

Additionally, information has been added to the Supporting Information at the end of the manuscript.

I have updated the version of the protocol on protocol.io; however, I was unable to add the custom code, manuscript title, or manuscript number during the update.

“TK:

JSPS KAKENHI Grant Number JP19K09100

Japan Society of Promotion of Science(JSPS)

https://www.jsps.go.jp”

Response to journal

I have included the statement, "The funders had no role in study design, data collection and analysis, decision to publish, or preparation of the manuscript." in the cover letter, as the funders had no involvement. Please update the online submission accordingly.

“This work was supported by JSPS KAKENHI Grant Number JP19K09100”

“TK:

JSPS KAKENHI Grant Number JP19K09100

Japan Society of Promotion of Science(JSPS)

https://www.jsps.go.jp”

Response to Journal

As per your instructions, I have removed the funding information from the acknowledgements section.

6. We note that Figures 3, 4, 5 and 6 in your submission contain copyrighted images. All PLOS content is published under the Creative Commons Attribution License (CC BY 4.0), which means that the manuscript, images, and Supporting Information files will be freely available online, and any third party is permitted to access, download, copy, distribute, and use these materials in any way, even commercially, with proper attribution. For more information, see our copyright guidelines: http://journals.plos.org/plosone/s/licenses-and-copyright.

a. You may seek permission from the original copyright holder of Figures 3, 4, 5 and 6 to publish the content specifically under the CC BY 4.0 license.

Response to Journal

Since the copyright holders of Fig3, Fig4, Fig5, Fig6, Fig7, S_Fig1 are the authors themselves, we have created and uploaded an original proof of permission instead of using the recommended Permission Form. If this approach is inappropriate, please provide further instructions.

Response to Journal

As per your instructions, we have added captions for the Supporting Information files at the end of the manuscript.

Response to Reviewer #1

Reviewer #1: Following a thorough review of the manuscript, the following issues and problems were identified. By addressing these issues, it is anticipated that the paper will exhibit enhanced quality and reliability, thereby more clearly demonstrating the value of the proposed system.

Response to Reviewer #1

We appreciate the review of our manuscript and the valuable comments that have contributed to its improvement.

Based on the feedback received, we revised the content to remove excessive evaluation and reorganized the discussion. The main achievement of this study is the development of a system that imports organ surface data, obtained through deep learning-based automatic extraction techniques such as 3D Slicer, into CAD software. This allows for modifications based on the original image data to create a more ideal representation.

The data generated by this system is expected to be applicable to various surgical simulations, such as surgical planning and training. However, since no evaluation of the simulator has been conducted, we limit our discussion to the potential applications.

Below, we provide our revisions and responses to the comments received.

1. The methodology is inadequate in terms of detail.- The automatic segmentation process and the deep learning model used are not clearly explained.

In order to enhance reproducibility, it is essential to provide more specific information, including the procedure, the software used, and the parameter settings.

Response to Reviewer #1

We have published the overall workflow of the protocol on protocols.io, while the Python scripts required for its implementation have been uploaded to a GitHub repository making them easily accessible via a link from protocols.io.

Additionally, to enable readers to replicate this protocol using specific TCIA data, we have provided a detailed demonstration on GitHub. This demonstration covers the entire process—from obtaining data from TCIA to setting up the simulation environment in Blender (corresponding to Fig. 3C in the manuscript)—including parameter settings and other necessary configurations.

https://dx.doi.org/10.17504/protocols.io.bp2l6d19dvqe/v4

https://github.com/tk1971-Jpn/Demonstration-of-Organ-data-set-creation

2. Lack of evaluation metrics- Objective metrics are absent, hindering the evaluation of the effectiveness of the proposed system. For instance, the anatomical accuracy of the generated model and quantitative evaluation of training effects are required.

Response to Reviewer #1

Thank you for pointing out this important aspect. To evaluate the usefulness of this protocol, we obtained data from 12 different projects of The Cancer Genome Atlas (TCGA) within the TCIA dataset, each representing a different type of cancer. We measured the processing time for each step and included the data in S2 File and S1 Fig in the manuscript.

The time required to process the data—from DICOM files to setting up the simulation environment in Blender (corresponding to Fig. 3C in the manuscript)—was approximately 7 minutes on average when using 3D Slicer's Fast mode, which produces a slightly rough surface. When using the Full-resolution mode, which provides more detailed data, the processing time was around 22 minutes. Even when including the time required for launching the program and configuring parameters, the entire process can be completed within 15 to 30 minutes.

Assessing the accuracy of the surface data obtained from 3D Slicer is beyond the scope of this study, and due to the lack of a gold standard dataset, it is not possible to quantitatively evaluate its precision. However, by refining the extracted surface data within the 3D space using the original images as a reference, we believe that it is possible to obtain even more precise data compared to the automatically extracted models.

(p.11, line196)

3. The number of samples and diversity (age, gender, body shape, etc.) of the CT data used are not clearly stated. These factors are of particular relevance given their impact on the generalisability of results.

Response to Reviewer #1

Since patient body size data is not publicly available in TCIA, we were unable to obtain it. However, as shown in S1 Fig, we believe that this system has been demonstrated to function effectively regardless of body size, disease type, original image resolution, or dataset size.

4. Specific solutions to AR issuesThe issue of spatial consistency in AR is mentioned, but no specific approach to solving this problem or future research policy is presented.

Response to Reviewer #1

The resolution of issues related to AR simulation training is not the focus of this study. Therefore, we intend to mention AR only as one of the potential applications of this system.

(p.15, line278)

(p.18, line356)

5. Verification of cost-effectiveness and comparison with other existing methodsThe assertion that the system is "cost-effective" is not supported by specific evidence or comparative data.

A comparative analysis between the proposed system and other extant surgical training methods is absent. This would allow for a more robust demonstration of the system's superiority.

In addition, the long-term effectiveness of the system merits further investigation. The present paper does not examine the long-term effects of training using this system or its impact on actual surgical outcomes.

Response to Reviewer #1

As you pointed out, it is inappropriate to describe this as a cost-effective system since a direct comparison with existing simulation systems is not possible. However, this system is unique in that it allows surface data extracted from a DICOM viewer to be further processed within CAD software, which has not been achieved before.

The construction of simulation systems typically requires significant costs and time for data processing and subsequent simulator design. This system enables healthcare professionals to complete these steps using only free software, potentially reducing costs. Therefore, we will limit our discussion to the possibility of cost reduction.

(p.18, line362)

Response to Reviewer #2

Reviewer #2: General Comments

Thank you for the opportunity to review “Development of Anatomically Accurate Digital Organ Models for Surgical Simulation and Training – A Methodology for Reconstructing and Applying 3D Organ Datasets in Real Space”. Overall, I think this is a worthwhile manuscript. I have a number of edits, suggestions, and comments that I would like the authors to consider before this manuscript is ready for publication.

Response to Reviewer #2

Thank you for reviewing our manuscript and for providing valuable comments that have helped improve it.

Based on the feedback received, we revised the content to remove excessive evaluation and reorganized the discussion. The main achievement of this study is the development of a system that imports organ surface data, obtained through deep learning-based automatic extraction techniques such as 3D Slicer, into CAD software. This allows for modifications based on the original image data to create a more ideal representation.

The data generated by this system is expected to be applicable to various surgical simulation

---

## [Decision Letter · Decision Letter 1]

25 Feb 2025

Development of Anatomically Accurate Digital Organ Models for Surgical Simulation and Training

PONE-D-24-54417R1

Dear Dr. Kimura,

We’re pleased to inform you that your manuscript has been judged scientifically suitable for publication and will be formally accepted for publication once it meets all outstanding technical requirements.

Kind regards,

James J Cray Jr., Ph.D.

Academic Editor

PLOS ONE

Additional Editor Comments (optional):

Reviewers' comments:

Reviewer's Responses to Questions

**Comments to the Author**



Reviewer #1: Yes

Reviewer #2: Yes

2. Has the protocol been described in sufficient detail?

To answer this question, please click the link to protocols.io in the Materials and Methods section of the manuscript (if a link has been provided) or consult the step-by-step protocol in the Supporting Information files.

Reviewer #1: Yes

Reviewer #2: Yes

3. Does the protocol describe a validated method?

Reviewer #1: Yes

Reviewer #2: Yes

4. If the manuscript contains new data, have the authors made this data fully available?

Reviewer #1: Yes

Reviewer #2: N/A

**5. Is the article presented in an intelligible fashion and written in standard English?**

Reviewer #1: Yes

Reviewer #2: Yes

Reviewer #1: The revised paper was judged to have reached a level of scientific merit that adequately addresses the points raised in the peer review. In particular, the main strength is the improved methodological transparency provided by the publication of the protocol (DOI: 10.17504/protocols.io.bp2l6d19dvqe/v4) and the GitHub code sharing (https://github.com/tk1971-Jpn). The technical originality of the specific example of the laparoscopic cholecystectomy simulator (Figure 6) and the mechanism for reproducing the metal base coordinate system using Blender-Python integration (Figure 5) is high. However, the following areas for improvement remain.

#1 Main issues of concern

1) Inadequate verification of data accuracy

Inter-observer variability in the process of modifying automatically extracted data has not been evaluated. Comparison of cube coefficients and statistical analysis of surface shape by three or more surgeons is required. Quantitative comparative data, such as mean surface distance of the liver margin, are lacking for the difference in accuracy of automatic segmentation between Horos and 3D Slicer.

2) AR projection accuracy issues

Quantitative data on positional errors between virtual and real objects (e.g. verification of 5mm grid pattern projection) have not been measured. Measurement of projection errors in millimetres is essential to ensure accuracy of 1.5mm or less, which is required for laparoscopic surgery training.

3) Technical significance

It is worth noting that the paper clearly defines a method for removing unnecessary parts from existing organ data (subtractive approach) and aims to improve the efficiency of simulator design (p.18, line360) [1]. However, there is a lack of comparative studies with automatic segmentation methods other than deep learning (nnU-Net) (e.g. atlas-based), and clarification of the technical limitations is needed.

#2 Future perspectives

It is necessary to systematically evaluate the impact of different CT scanning protocols (tube voltage 80-140kVp, reconstruction algorithm FBP vs. IR) on data quality. The versatility of the software should be enhanced by adding examples of analysis of data from the user's own institution in addition to the TCIA data.

It is also hoped to implement tissue hardness simulation (Young's modulus range 10-100kPa) using the Blender physics engine.

#3 Conclusion

The response to the peer review comments is adequate and we recommend publication at this stage. The remaining issues are considered to be within an acceptable range, indicating the possibility of future development. The next important step towards practical application in education will be the quantitative evaluation of the training effect (rate of reduction in operative time, rate of reduction in complications). The academic significance is high as a model that combines the democratising effect of open source infrastructure and medical AI clinical knowledge. In conclusion, the revised manuscript adequately addresses the reviewers' comments and is recommended for publication at this time. The remaining issues are considered to be within an acceptable range, indicating the possibility of future development. The next important step will be to quantitatively evaluate the training effects (reduction in operative time and complication rate) for practical application in the educational field.

Reviewer #2: I want to thank the authors for addressing all of the comments I made previously. I believe that this revised manuscript is much stronger and is ready for acceptance.

**Do you want your identity to be public for this peer review?** For information about this choice, including consent withdrawal, please see our Privacy Policy

Reviewer #1: **Yes: ** Takashi Kamio

Reviewer #2: No

---

## [Editor Report · Acceptance letter]

PONE-D-24-54417R1

PLOS ONE

Dear Dr. Kimura,

I'm pleased to inform you that your manuscript has been deemed suitable for publication in PLOS ONE. Congratulations! Your manuscript is now being handed over to our production team.

Kind regards,

on behalf of

Dr. James J Cray Jr.

Academic Editor

PLOS ONE